# Molecular Biological and Clinical Understanding of the Statin Residual Cardiovascular Disease Risk and Peroxisome Proliferator-Activated Receptor Alpha Agonists and Ezetimibe for Its Treatment

**DOI:** 10.3390/ijms23073418

**Published:** 2022-03-22

**Authors:** Hidekatsu Yanai, Hiroki Adachi, Mariko Hakoshima, Hisayuki Katsuyama

**Affiliations:** Department of Diabetes, Endocrinology and Metabolism, National Center for Global Health and Medicine Kohnodai Hospital, 1-7-1 Kohnodai, Chiba 272-8516, Japan; dadachidm@hospk.ncgm.go.jp (H.A.); d-hakoshima@hospk.ncgm.go.jp (M.H.); d-katsuyama@hospk.ncgm.go.jp (H.K.)

**Keywords:** cardiovascular diseases, ezetimibe, peroxisome proliferator-activated receptor alpha, statins, triglyceride

## Abstract

Several randomized, double blind, placebo-controlled trials (RCTs) have demonstrated that low-density lipoprotein cholesterol (LDL-C) lowering by using statins, including high-doses of strong statins, reduced the development of cardiovascular disease (CVD). However, among the eight RCTs which investigated the effect of statins vs. placebos on the development of CVD, 56–79% of patients had the residual CVD risk after the trials. In three RCTs which investigated the effect of a high dose vs. a usual dose of statins on the development of CVD, 78–87% of patients in the high-dose statin arms still had the CVD residual risk after the trials. An analysis of the characteristics of patients in the RCTs suggests that elevated triglyceride (TG) and reduced high-density lipoprotein cholesterol (HDL-C), the existence of obesity/insulin resistance, and diabetes may be important metabolic factors which determine the statin residual CVD risk. To understand the association between lipid abnormalities and the development of atherosclerosis, we show the profile of lipoproteins and their normal metabolism, and the molecular and biological mechanisms for the development of atherosclerosis by high TG and/or low HDL-C in insulin resistance. The molecular biological mechanisms for the statin residual CVD risk include an increase of atherogenic lipoproteins such as small dense LDL and remnants, vascular injury and remodeling by inflammatory cytokines, and disturbed reverse cholesterol transport. Peroxisome proliferator-activated receptor alpha (PPARα) agonists improve atherogenic lipoproteins, reverse the cholesterol transport system, and also have vascular protective effects, such as an anti-inflammatory effect and the reduction of the oxidative state. Ezetimibe, an inhibitor of intestinal cholesterol absorption, also improves TG and HDL-C, and reduces intestinal cholesterol absorption and serum plant sterols, which are increased by statins and are atherogenic, possibly contributing to reduce the statin residual CVD risk.

## 1. Introduction

A great number of epidemiological studies, including the Framingham study, have shown that an elevation of serum low-density lipoprotein cholesterol (LDL-C) increases the incidence and death from cardiovascular disease (CVD), such as coronary heart disease (CHD). The 4S, a randomized trial of cholesterol lowering in 4444 patients with CHD, reported in 1994, showed that the decrease in LDL-C by using simvastatin made a significant reduction of CVD by 35% [1]. The WOSCOPS, which studied the primary prevention of CVD by statin, showed that pravastatin lowered LDL-C levels by 26%, and reduced coronary events (nonfatal myocardial infarction (MI) or death from CHD) by 31% in the pravastatin group [2]. After the publication of the WOSCOPS, the ASCOT-LLA [3], AFCAPS/TexCAPS [4], CARDS [5], and JUPITER [6] studies showed a primary prevention of CVD by statins. In addition to the 4S, the LIPID [7] and CARE [8] studies demonstrated a secondary prevention of CVD by statins. 

In the 2000s, strong statins showing an intense lowering effect of LDL-C appeared, and randomized, double blind, placebo-controlled trials (RCTs) to study the effect of high doses and usual doses of statins on the development of cardiovascular events have been performed. The PROVE IT-TIMI 22 study allocated patients with acute coronary syndrome (ACS) to intensive (daily 80 mg of atorvastatin) and moderate (daily 40 mg of pravastatin) lipid-lowering therapy [9]. An intensive lipid-lowering therapy using atorvastatin decreased LDL-C by 51% and reduced cardiovascular events by 16%. This study demonstrated that, in patients who have recently had ACS, an intensive lipid-lowering statin therapy induced a greater protection against death or major cardiovascular events as compared with a standard therapy, indicating that ACS patients obtain a benefit from the early and continuous lowering of LDL-C to levels substantially below the previous target level. In the IDEAL study, patients with a previous MI were randomly assigned to receive a high dose of atorvastatin (80 mg/day) or a usual dose of simvastatin (20 mg/day) [10]. The mean LDL-C levels were 104 mg/dL in the simvastatin group and 81 mg/dL in the atorvastatin group. Major cardiovascular events occurred in 608 patients in the simvastatin group and 533 patients in in the atorvastatin group (hazard ratio (HR), 0.87; 95% confidence interval (CI); 0.77–0.98; *p* = 0.02). In the TNT study, the efficacy and safety of lowering the LDL-C levels below 100 mg/dL were evaluated in patients with CHD [11]. Patients were randomly assigned to a double-blind therapy and received either daily 10 mg or 80 mg of atorvastatin. The LDL-C levels were 77 mg/dL during treatment with 80 mg atorvastatin and 101 mg/dL during treatment with 10 mg atorvastatin. The HR for an absolute reduction in the rate of major cardiovascular events was 0.78, proposing an intensive lipid-lowering therapy with daily 80 mg of atorvastatin in patients with CHD provides a significant clinical benefit beyond that afforded by the treatment with daily 10 mg of atorvastatin. 

The Cholesterol Treatment Trialists’ (CTT) Collaboration performed a prospective meta-analysis of data from 90,056 individuals in 14 RCTs of statins [12]. There was a 12% proportional reduction in all-cause mortality and a 21% proportional reduction in major vascular event per mmol/L reduction in LDL-C. The proportional reduction in major vascular events differed significantly according to the absolute reduction in LDL-C achieved. These results emphasized the need to consider prolonged statin treatment with substantial LDL-C reduction in patients at high CVD risk. Furthermore, the CTT undertook meta-analyses of individual participant data from RCTs involving more versus less intensive statin regimens (five trials) and of statin versus control (twenty-one trials) [13]. When statins deceased LDL-C by 1.0 mmol/L (38.7 mg/dL), reductions in major vascular events were obtained by 22%. Further, all-cause mortality was reduced by 10% by a 1.0 mmol/L LDL reduction, largely reflecting the significant reductions in deaths due to CHD and other cardiac causes. For LDL-C, “the lower, the better” was suggested to be correct to reduce the risk of cardiovascular events.

Therefore, the current treatment guidelines focus on LDL-C by using statins for reducing CVD risk [14,15,16,17,18]. However, the PROVE IT-TIMI 22 study and the TNT study showed that CVD events occurred in a substantial number of patients, despite having LDL-C levels below 80 mg/dL [9,11]. These recent studies have shown that the risk of development of CVD remains even when statins are used to strongly reduce LDL-C, and such a risk has become to be called the statin residual CVD risk. 

## 2. Properties and Normal Metabolism of Lipoproteins

To understand the association between lipid abnormalities and the development of atherosclerosis, which induces CVD, first we show the profile of the lipoproteins and their normal metabolism.

Cholesterol and TG are insoluble in water; therefore, these lipids must be transported in association with apoproteins. Lipoproteins are complex particles with a central core containing cholesterol esters (CE) and TG surrounded by free cholesterol (FC), phospholipids, and apoproteins. The compositions and apoproteins included in each lipoprotein are shown in Figure 1. The metabolism of the lipoproteins is shown in Figure 2. 

Chylomicron (CM) is the largest and lowest density lipoprotein synthesized and secreted from the small intestine after the ingestion of dietary fat. More than 80% of the composition of CM is TG. The main apoprotein of CM is apo B48, which is synthesized by the small intestine. After the release from the intestine into the lymphatic system, newly secreted CM enter the systemic circulation via the thoracic duct. CM obtains apo C-II and apo C-III from HDL in plasma. TG in CM is hydrolyzed by lipoprotein lipase (LPL), which requires apo C-II to have a normal function. TG is hydrolyzed to free fatty acid (FFA), which is used in skeletal and cardiac muscles as an energy source and/or is stored in adipose tissue in the form of TG. Reduction of TG in CM by LPL makes the CM remnant, which is taken up by the remnant receptor via apo E in the liver. 

VLDL has apo B100, which is synthesized by the liver. Approximately 55% of VLDL is TG. Once released from the liver, TG is carried in VLDL and is metabolized in the muscle and adipose tissue by LPL, releasing FFA and resulting in IDL formation. IDL are metabolized to LDL, which are taken up by the LDL receptor in tissues, including the liver. The proportion of TG in IDL and LDL are 24% and 12%, respectively, and the proportion of cholesterol in IDL and LDL are 29% and 37%, respectively. CM, CM remnant, VLDL, and IDL are classified as TG-rich lipoproteins. Approximately 25–75% of VLDL remnants are not uptaken by the liver, but rather are rather changed to LDL [19]. 

CM is an extrinsic lipoprotein, whereas VLDL, IDL, and LDL are endogenous lipoproteins. VLDL, IDL, and LDL have apo B-100; therefore, such lipoproteins are called apo B-100-containing lipoproteins.

The reverse cholesterol transport by HDL is shown in Figure 3. Apo AI, the main apoprotein of HDL is produced in the liver and the small intestine and receives FC via the ATP-binding cassette transporter A1 (ABCA1), present on the surface of the cell membranes of foam cells, and becomes primitive HDL, preβ-HDL. This process is called cholesterol efflux. ABCG1, another type of ABC transporter, is involved in cholesterol efflux and mediated by HDL2 and HDL3, which are relatively lipid-rich, rather than lipid-free apo AI, unlike ABCA1. In addition to the ABC transporter-mediated pathway, the lecithin–cholesterol acyltransferase (LCAT) lowers the FC concentration on the HDL surface and forms a concentration gradient, which is cholesterol efflux by passive diffusion. As the reverse cholesterol transfer mechanism after cholesterol efflux, there is a pathway mediated by SR-BI and a pathway mediated by apo B-containing lipoproteins. In the former pathway, mature HDL is taken up by the liver via SR-BI. SR-BI binds HDL, but HDL itself is not incorporated and only CE contained in HDL is selectively incorporated into the liver. In the latter pathway, CE in HDL is transferred by CETP to VLDL, IDL, and LDL, which is then taken up by the liver via the LDL receptor and metabolized to bile acids.

## 3. What Is the Statin Residual CVD Risk?

In addition to elevated LDL-C, hypertriglyceridemia and reduced HDL-C are also dyslipidemia involved in the development of arteriosclerosis. In Japan, hypertriglyceridemia and reduced HDL-C were defined as fasting TG > 150 mg/dL and HDL-C < 40 mg/dL, respectively. Such dyslipidemia was commonly observed in the metabolic syndrome; therefore, it was included in the diagnostic criteria for the metabolic syndrome in Japan. 

The meta-analysis to investigate the association between the treatment-induced change in HDL-C and total death, CHD death, and the events adjusted for changes in the LDL-C and drug classes in the RCTs of lipid-modifying interventions was performed [20]. Available data suggest that simply increasing the amount of circulating HDL-C does not reduce the risk of CHD events, CHD deaths, or total deaths. The presence of accompanying hypertriglyceridemia and insulin resistance may be more important for the development of CVD than the HDL value.

In the EMPATHY study, 5042 patients with hypercholesterolemia with diabetic retinopathy, who had no history of CHD, were allocated to the intensive lipid-lowering group (LDL-C < 70 mg/dL) and the usual lipid-lowering group (100 mg/dL ≤ LDL-C < 120 mg/dL) [21]. In the subanalysis of the EMPATHY study, the HR of the fourth quartile of TG (135–185 mg/dL) for the development of CVD was 1.65 (95% CI, 1.02–2.65; *p* = 0.0331) [22], suggesting that the serum TG level of 135 can be the residual CVD risk. The study examined the relationship of fasting TG levels to outcomes after ACS in patients treated with statins. Long-term and short-term relationships of TG to risk after ACS were examined in the dal-OUTCOMES trial and the atorvastatin arm of the MIRACL (Myocardial Ischemia Reduction with Acute Cholesterol Lowering) trial, respectively [23]. Among the patients with ACS treated effectively with statins, fasting TG predicted long-term and short-term cardiovascular risk. Interestingly, the risk for the development of the recurrence of CVD during a short period after ACS was significantly increased in the quartile with the TG of 135–195 mg/dL as compared with the lowest quartile (MIRACLE), and the risk during a long period after ACS was significantly increased in the quartile with the TG of 103–130 mg/dL as compared with the lowest quartile (dal-OCOMES) [23], indicating that the serum TG level of 103 can be the residual CVD risk. 

To understand the statin residual metabolic CVD risk, we evaluated body mass index (BMI) at baseline and serum TG and HDL-C levels achieved after the trials in eight RCTs which investigated the effect of statins on the development of CVD (Table 1) and three RCTs which investigated the effect of the high dose and usual dose of statins on the development of CVD (Table 2). A BMI > 25 kg/m^2^ and the prevalence of diabetes > 10% at baseline, and HDL-C < 40 mg/dL after the trials, were considered to be the residual metabolic CVD risk. According to the result of the dal-OCOMES [23], a serum TG > 103 mg/dL was considered to be the residual metabolic CVD risk.

Among the eight RCTs which investigated the effect of statins on the development of CVD, 56–79% of patients had the residual CVD risk after the trials. In these studies, the annual residual CVD risk was surprisingly high, with 76.8–95.2%. In the three RCTs which investigated the effect of the high dose vs. usual dose of statins on the development of CVD, 78–87% of patients in the high-dose statin arms still had the CVD residual risk after the trials. In these studies, the annual residual CVD risk was extremely high, with 92–97.3%.

Serum HDL-C < 40 mg/dL and TG > 103 mg/dL after the trials were observed in approximately 18% and 82% of RCTs, and a BMI > 25 kg/m^2^ at baseline was observed in all RCTs. The prevalence of diabetes was over 10% in three of eight RCTs which investigated the effect of statins on the development of CVD (37.5%), and the prevalence of diabetes was over 10% in all RCTs which investigated the effect of the high dose vs. usual dose of statins on the development of CVD. Atherogenic dyslipidemia, such as elevated TG and reduced HDL-C, diabetes, and insulin resistance, which are induced by obesity, may be important metabolic factors which determine the statin residual CVD risk [24,25]. 

## 4. Molecular and Biological Mechanisms for the Development of Atherosclerosis by High TG and/or Low HDL-C in Insulin Resistance 

Insulin resistance enhances the expression and activity of hormone-sensitive lipase (HSL) in adipose tissue. HSL catalyzes the hydrolysis of TG into FFA and glycerol (Figure 4) [26]. Further, insulin resistance enhances intestinal cholesterol absorption, attributable to changes in Niemann–Pick C1-Like 1 (NPC1L1) [27]. Intestinal cholesterol absorption in obese people is more than twice that in nonobese people [28]. Therefore, FFA entry from the adipose tissue to the liver, and cholesterol entry from the intestine to the liver, are enhanced by insulin resistance, which leads to the overproduction of VLDL. 

Since insulin promotes apo B100 degradation, insulin resistance is associated with reduced apo B100 degradation [29]. Insulin resistance also enhances the expression of the microsomal TG transfer protein (MTP), which plays a crucial role in the assembly of VLDL [30]. Insulin resistance is associated with elevated hepatic apo C-III production [31]. Insulin resistance decreases the activity of LPL, which catabolizes TG-rich lipoproteins [32]. The synthesis of HDL is closely related with the catabolism of TG-rich lipoproteins by LPL [33]. In insulin resistance, TG-rich lipoproteins such as VLDL, IDL, and CM remnants increase, and HDL decreases due to reduced LPL activity. The activity of hepatic TG lipase (HTGL), which facilitates HDL catabolism, is associated with insulin resistance [34]. An increased rate of clearance by HTGL may be associated with low serum HDL-C [34]. HTGL activity influences LDL size and buoyancy [35], and patients with high HTGL have smaller, denser LDL particles as compared with subjects with low HTGL activity [36]. Increased HTGL activity due to insulin resistance may enhance the formation of small dense LDL.

Two major physically distinct species of VLDL exist: larger VLDL1 (50–80 nm diameter, 70% TG mass) and smaller VLDL2 (30–50 nm diameter, 30% TG mass) [37]. At normal TG concentrations, VLDL1 and VLDL2 circulate in approximately equal proportions. Hepatic TG accumulation and insulin resistance influence VLDL1 secretion, including the hepatic TG stores and insulin resistance [38,39]. LDL derived from VLDL1 was also shown to have a slower clearance rate and a longer duration to decay, inducing the formation of small dense LDL [40,41]. The increased atherogenic potential of small dense LDL is suggested by the greater propensity for transport into the subendothelial space, the increased binding to arterial proteoglycans, and the susceptibility to oxidative modification [42]. Small dense LDL are cleared more slowly from plasma than LDL due to the compositional or structural features of small dense LDL that lead to retarded clearance [43].

Remnants have undergone extensive intravascular remodeling. LPL, HTGL, and CETP induce structural and atherogenic changes that distinguish remnants from nonremnant lipoproteins [37]. Via the rapid LPL-mediated removal of TG and the CETP-mediated exchange of TG for cholesterol from LDL and HDL, the remnant particles contain more cholesterol than nascent CM or VLDL [44]. Compared with CM or VLDL, remnants lose apo C-III and become enriched in apo E [37]. While CM and VLDL are prohibited from transcytosis by virtue of their size, the remnants can and do penetrate the artery wall [45,46,47]. Remnants efflux from the subendothelial space very slowly compared to native LDL, and therefore encounter increased internalization by macrophages [48,49]. Distinct from LDL, which is deficient in apo E and requires oxidation for uptake, remnants do not need oxidation to facilitate the accumulation in macrophages due to their apo E enrichment.

Because adipose tissues are the major sources of cytokines, increased adipose tissue mass is associated with the overexpression of tumor necrosis factor-alpha (TNF-α) and interleukin-6 (IL-6), and the underexpression of adiponectin in adipose tissue. The inflammation caused by TNF-α triggers the activation of mitogen-activated protein kinases [50]. Moreover, these kinases induce an inflammatory response through the activation of the transcription factors nuclear factor kappa B (NF-κB) and activator protein 1 (AP-1), which in turn promotes inflammatory gene expression [51,52].

The liver synthesizes acute phase reactants such as C-reactive protein (CRP), fibrinogen, and serum amyloid A (SAA), which are correlated with the development of CVD [53,54]. The proinflammatory status provides a potential link between insulin resistance and endothelial dysfunction [55]. Insulin resistance and its resulting elevation of TG-rich lipoproteins enhance the expression of plasminogen activator inhibitor-1 (PAI-1), which is the fibrinolytic inhibitor, in the liver and endothelial cells [56]. The von Willebrand factor (VWF) is also elevated in insulin-resistant states [56]. The vascular endothelium influences the vascular integrity, including the vessels themselves, platelets, clotting, and fibrinolytic systems, inflammation, and tissue repair [57,58]. Therefore, an endothelial injury by inflammation induces vasoconstriction, platelet and leukocyte activation, and adhesion by the upregulation of VWF and platelet-activating factor (PAF), the promotion of thrombin formation, and coagulation and fibrin deposition at the vascular wall by the upregulation of tissue factor (TF) and PAI-1 [58,59]. Smooth muscle cells and endothelial cells play active roles in inflammatory reactions [60]. Several cytokines have been found to influence the growth of smooth muscle cells [60]. The vessel wall tone is regulated by signals from endothelial cells. Injured endothelial cells release endothelin-1 (ET-1), which promotes blood vessel constriction. ET-1 also induces smooth muscle cell proliferation [61].

Early atherogenesis is characterized by the recruitment of inflammatory cells to an injured endothelium. Inflammatory cells are attached to injured endothelial cells by adhesion molecules such as the vascular cell adhesion molecule-1 (VCAM-1) and transmigrates into the intima [62]. VCAM-1 expression is induced by proinflammatory cytokines such as TNFα, IL-1, and IL-6. Such responses may lead to the formation of the plaques prone to rupture [61]. Therefore, VCAM-1 levels are correlated with cardiovascular risk.

## 5. Current Views on Estimation of Residual Cardiovascular Risk in Patients Treated with Statins 

In the guidelines on the management of CVD, LDL-C remains the primary target while apo B and non-HDL-C can be secondary targets. Atherogenic TG-rich lipoproteins such VLDL and IDL, and LDL, have apo B-100. The serum level of non-HDL-C is calculated by subtracting HDL-C from total cholesterol; therefore, non-HDL also includes atherogenic TG-rich lipoproteins (VLDL, IDL and remnant) and LDL. Johannesen CDL et al. studied to determine if elevated apo B and/or non-HDL-C are superior to elevated LDL-C in identifying statin-treated patients at residual risk of all-cause mortality and MI. In total, 13,015 statin-treated patients from the Copenhagen General Population Study were included, with 8 years median follow-up. High levels of apo B and non-HDL-C were associated with an increased risk of all-cause mortality and MI, whereas no such associations were found for high LDL-C, suggesting that elevated apo B and non-HDL-C, but not LDL-C, are associated with residual risk of all-cause mortality and MI in statin-treated patients [63]. 

## 6. Current Views on Clinical Approaches Reducing the Statin Residual CVD Risk 

To reduce the statin residual CVD risk, the following approaches should be needed: (1) the intensification of statin therapy with the aim to reach target values; (2) the improvement of lifestyle; (3) the incorporation of peroxisome proliferator-activated receptor alpha (PPARα) agonists and/or NPC1L1 inhibitor (ezetimibe), in particular in clinical scenarios such as high TG, low HDL-C, obesity, and metabolic syndrome. 

## 7. Molecular and Biological Mechanisms for Vascular Protection by PPARα Agonists

The most important class of medications to manage dyslipidemia due to insulin resistance can be PPARα agonists such as fibrates, because PPARα agonists induce a greater reduction of TG and a greater increase of HDL-C [64]. Therefore, we show the possible vascular protective mechanisms by PPARα agonists.

### 7.1. Properties of PPARs

PPARs, which belong to the superfamily of steroid–thyroid–retinoid nuclear receptors [65], are transcription factors activated by specific ligands and play an important role during cell signaling. PPARs participate in the regulation of the lipid metabolism, inflammation, and the development of atherosclerosis or diabetes. PPARs have three isoforms: PPARα, PPARγ, and PPARβ/δ [66]. PPARα is abundant in energy-demanding tissues, such as the liver, kidney, and muscles; PPARγ is predominantly found in adipose tissue, macrophages, and the large intestine; PPARβ/δ is ubiquitously distributed [67,68]. 

### 7.2. Effects of PPARα on Adipose Tissue

Altered properties of white adipose tissue (WAT) by obesity are associated with inflammation, insulin resistance, enhanced lipolysis, ectopic lipid accumulation, and reduced energy expenditure [69,70]. Brown fat expansion and/or activation increases energy expenditure and induces a negative energy balance [71] (Figure 5). Brown fat can utilize blood glucose and lipids, and results in improved glucose metabolism and serum lipids, independent of weight loss [71]. PPARα agonists can induce the browning of WAT [72]. The WAT altered by browning produces less cytokines, which may improve systemic insulin resistance and inflammation, and may result in the reduction of FA release from the adipose tissue. Further, PPARα agonists enhance adiponectin production [73], which may be also beneficially associated with systemic insulin resistance and inflammation [74].

### 7.3. Effects of PPARα on Skeletal Muscle

The activation of PPARα markedly stimulated the muscle expression of two key enzymes involved in FA oxidation, carnitine palmitoyl transferase and acyl-CoA oxidase [75] (Figure 5). Moreover, the liver and muscle tissue TG content were significantly reduced by the PPARα treatment [75]. Elevated adiponectin induces the upregulation of the VLDL receptor (VLDL-R) in the skeletal muscle. An increase of VLDL-R expression has been observed in adiponectin-treated myotubes, leading to the increased VLDL catabolism [76]. 

### 7.4. Effects of PPARα on TG-rich Lipoproteins

Elevated FA oxidation in the skeletal muscle and the reduced FA release from the adipose tissue by PPARα agonists decrease FA entry to the liver and may result in the reduction of hepatic VLDL production. PPARα agonists reduce hepatic TG synthesis by decreasing apo C-III production [64] (Figure 5). Further, the treatment with the PPARα agonist simulated the expression of enzymes involved in FA oxidation, leading to a concomitant decrease of hepatic TG levels [77]. PPARα agonistic activity reduces serum TG by lowering hepatic VLDL production and by elevated FA oxidation. PPARα agonists stimulate the activity of LPL, which further reduce VLDL [64]. As a result, there is an increase in HDL levels and a decrease in small dense LDL [78].

Fenofibrate reduced the postprandial increase of CM remnants in patients with combined hyperlipidemia [79]. Furthermore, fenofibrate remarkably suppressed the postprandial increase of TG and apo B48 by reducing CM production in the intestine [80,81]. 

### 7.5. Effects of PPARα on Reverse Cholesterol Transport

Fibrates activate PPARα and elevate HDL-C levels via the transcriptional induction of apo AI and apo AII formation [64] (Figure 5). In addition, PPARα regulates the HDL metabolism by promoting HDL-mediated cholesterol efflux from macrophages via the enhanced expression of ABCA1 [82]. 

### 7.6. Effects of PPARα on Vascular Integrity

#### 7.6.1. Effects on Inflammation

PPARα may have a direct effect on inflammation and atherosclerosis through the modification of the transcription factors NF-κB and AP-1 [83] (Figure 6). PPARα agonists decrease plasma CRP, TNFα, IL-6 levels, and the IL-1-induced expression of CRP and IL-6 [84,85,86,87]. 

The meta-analysis was conducted to assess the effects of fibrates on CRP concentrations, and the relationship between changes in CRP and lipid measures [88]. Compared with the placebo, treatment with fibrates significantly decreased CRP concentrations (weighted mean difference (WMD), −0.47 mg/L; 95%CI, −0.93 to −0.01 mg/L, *p* = 0.046). Fibrates significantly reduced CRP concentrations in trials with higher baseline CRP concentrations. There was a significant correlation between the change in CRP and the change in HDL-C.

An RCT of canakinumab, a therapeutic monoclonal antibody targeting interleukin-1β, involving 10,061 patients with previous MI and a high-sensitivity C-reactive (hs-CRP) level of 2 mg or more per liter was performed [89]. Canakinumab did not reduce lipid levels from baseline. At a median follow-up of 3.7 years, the 150 mg dose met the prespecified multiplicity-adjusted threshold for statistical significance for the primary endpoint and the secondary endpoint that additionally included hospitalization for unstable angina that led to urgent revascularization. Anti-inflammatory therapy targeting the interleukin-1β innate immunity pathway with canakinumab led to a significantly lower rate of recurrent cardiovascular events than the placebo, independent of lipid-level lowering. This result proposes inflammatory changes are responsible for the maintenance of the atherosclerotic process and may underlie vascular complications.

#### 7.6.2. Effects on Vasoconstriction and Smooth Muscle Cell Proliferation

ET-1 promotes blood vessel constriction. ET-1 induces the proliferation of smooth muscle cells. Such ET-1 responses are inhibited by nitric oxide (NO), which is released from the endothelial cells. PPARα activation inhibits the thrombin-mediated induction of ET-1 [90]. In the endothelial cells, PPARα agonists inhibited an induction of ET-1 release by oxidized LDL [91]. PPARα agonists were reported to enhance NO release [92,93]. Further, PPARα activation induces the cyclin-dependent kinase inhibitor p16, inhibiting smooth muscle cell proliferation [94]. Reduced smooth muscle cell proliferation may be associated with plaque stability. 

#### 7.6.3. Effects on Adhesion of Monocyte to Endothelial Cells

PPARα agonists inhibit the cytokines-mediated induction of the transcriptional expression of VCAM-1 [95,96], reducing the adhesion of the monocyte to endothelial cells. 

#### 7.6.4. Effects on Oxidized LDL Formation

An increased PPARα expression was correlated with the reduction of oxidized LDL in atherosclerotic plaques in insulin-resistant mice [97]. Further, PPARα agonists were associated with the reduction of LDL susceptibility to oxidation [98]. PPARα activation reduced the level of the reduced form of nicotinamide dinucleotide phosphate (NADPH) oxidase, which produces superoxide in endothelial cells [99]. PPARα agonists increase the expression of superoxide dismutase, which scavenge and process free radicals [100]. An inhibition of NFκB activation may be the mechanism that PPARα activators uses to repress superoxide-mediated atherogenesis [101].

#### 7.6.5. Effects on Procoagulant State

PPARα has also been reported to limit the expression of TF, a potent procoagulant protein, in human monocytes and macrophages [102,103]. 

The meta-analysis of head-to-head RCTs to compare the efficacy of statins and fibrates on plasma fibrinogen concentrations showed a significantly greater effect of fibrates vs. statins in lowering plasma fibrinogen concentrations (WMD, −40.7 mg/dL; 95% CI, −55.2, −26.3, *p* < 0.001), suggesting that fibrinogen metabolism might be responsible for the distinct effects of statins and fibrates in reducing cardiovascular events [104].

In the meta-analysis to analyze the risks and benefits of low-dose aspirin in patients with type 2 diabetes without cardiovascular conditions, the low-dose aspirin use was associated with the reduced risk for major adverse cardiovascular events (MACE) in the moderate/high-risk group (odds ratio (OR), 0.88; 95% CI, 0.80 to 0.97) [105]. Based on the within-subgroups results, the baseline cardiovascular risk does not modify the effect of aspirin therapy. This study suggests that antiplatelet activity can reduce CVD in the high-risk subjects, such as type 2 diabetic patients. 

## 8. Effects of PPARα Agonists on CVD

In the meta-analysis to investigate the influence of fibrates on vascular risk reduction in subjects with atherogenic dyslipidemia [106], compared to the placebo, the greatest benefit with the fibrate treatment was seen in high TG subjects, and the fibrate therapy reduced the risk of vascular events (relative risk (RR), 0.75; 95% CI, 0.65 to 0.86; *p* < 0.001), as well as in subjects with both high TG and low HDL-C (RR, 0.71; 95% CI, 0.62 to 0.82; *p* < 0.001). Less benefit was noted in low-HDL-C subjects (RR, 0.84; 95% CI, 0.77 to 0.91; *p* < 0.001). Among those subjects with neither high TG nor low HDL-C, fibrate therapy did not reduce subsequent vascular events.

Two major trials with fenofibrate in patients with type 2 diabetes have failed to reduce MACE. The Fenofibrate Intervention and Event Lowering in Diabetes (FIELD) study randomized diabetic patients with a TG of 154 mg/dL, HDL-C of 43 mg/dL, and LDL-C of 119 mg/dL to the placebo and fenofibrate over a 5-year study period and did not show a significant reduction in the primary combined endpoint for CVD between the study groups [107]. However, the post hoc analysis showed that patients with an elevated TG (>200 mg/dL) or low HDL-C (<40 mg/dL in men and <50 mg/d in women) derived greater CV risk reduction with fenofibrate [108]. The Action to Control Cardiovascular Risk in Diabetes Lipid trial (ACCORD Lipid) reported no effects on the primary composite outcome of fenofibrate added to simvastatin [109]. However, in a prespecified subgroup analysis, there was a trend towards the benefit of fenofibrate in patients with a TG level of ≥204 mg/dL or HDL-C of ≤34 mg/dL [109]. The post hoc analyses of the fibrate trials have shown reductions in CV events in subgroups with features of the metabolic syndrome, including overweight participants with high TG and low HDL-C [110,111]. 

The newly evolving evidence generally supports the value of fibrates in CV risk management. However, to confirm the usefulness of PPARα agonists for the statin residual CVD risk, well-designed studies focused on subjects with metabolic syndrome or atherogenic dyslipidemia in primary and secondary prevention should be needed in the future. 

## 9. The Significance of Cholesterol Absorption and Plant Sterol for the Statin Residual CVD Risk

Insulin resistance enhances intestinal cholesterol absorption attributable to changes in NPC1L1 [27]. Intestinal cholesterol absorption in obese people is more than twice that in nonobese people [28]. Diabetic patients had more NPC1L1 mRNA than the control subjects (*p* < 0.02) [112].

Miettinen, T.A., et al. studied changes in serum cholestanol and plant sterols (indexes of cholesterol absorption) and cholesterol precursors (indexes of cholesterol synthesis) in response to cholesterol reduction by way of 1 year of treatment with atorvastatin and simvastatin treatments in patients with CHD. Plant sterol concentrations were gradually increased by atorvastatin but decreased initially by simvastatin. Effective inhibition of cholesterol synthesis and the subsequent reduction in serum cholesterol levels by statins led to increases in serum plant sterol levels, probably as a result of reduced biliary secretion and the enhanced absorption of these sterols [113]. Because serum plant sterols have been claimed to be involved in the early development of atherosclerosis, the elevation of serum plant sterols can be the statin residual risk.

Miettinen, T.A., et al. studied to investigate whether the baseline serum cholestanol: cholesterol ratio, which is negatively related to cholesterol synthesis, could predict the reduction of coronary events in the 4S [114]. With the increasing baseline quarter of cholestanol distribution, the reduction in relative risk increased gradually from 0.623 (95% CI, 0.395 to 0.982) to 1.166 (0.791 to 1.72). The risk of recurrence of major coronary events increased 2.2-fold (*p* < 0.01) by multiple logistic regression analysis between the lowest and highest quarter of cholestanol. Measurement of serum cholestanol concentrations revealed a subgroup of patients with CHD in whom coronary events were not reduced by simvastatin treatment. This study suggests that cholesterol absorption can also be the statin residual risk. 

In mildly hypercholesterolemic type 2 diabetic patients, the only metabolic parameter differentiating CHD patients from non-CHD ones was significantly higher cholesterol absorption efficiency in the coronary patients, suggesting a significance of cholesterol absorption for atherogenesis [115]. 

## 10. Effects of Ezetimibe on the Statin Residual CVD Risk

Ezetimibe, an inhibitor of intestinal cholesterol absorption, can decrease LDL-C, TG, and apo B levels and increase HDL-C levels [116]. The meta-analysis showed that the combination of statins with ezetimibe as well as high-dose statin reduces plasma ox-LDL in comparison with low-to-moderate-intensity statin therapy alone [117]. Ezetimibe produced significant and progressive reductions in plasma plant sterol concentrations in patients with sitosterolemia [118]. The meta-analysis showed that ezetimibe/simvastatin produced significantly greater reductions compared with simvastatin alone in LDL-C (52.5% vs. 38.0%, respectively) and CRP levels (31.0% vs. 14.3%, respectively) [119]. Apart from being lipid-lowering, ezetimibe may exert certain off-target actions, such as being anti-inflammatory, antiatherogenic, and antioxidant, thus contributing to a further decrease of the CVD risk.

Ezetimibe treatment reduces the absorption of cholesterol and plant sterols. Ezetimibe treatment lowers LDL-C when added to the statin treatment. Ezetimibe treatment may reduce the statin residual CVD risk. In the IMPROVE-IT, 18,144 patients after ACS with the LDL-C from 50 to 125 mg/dL were randomized to 40 mg ezetimibe/simvastatin (E/S) or 40 mg placebo/simvastatin [120]. In patients with diabetes, E/S reduced the 7-year Kaplan–Meier primary endpoint event rate by 5.5%. The largest relative reductions in patients with diabetes were in MI (24%) and ischemic stroke (39%). Among patients without diabetes, those with a high-risk score experienced a significant (18%) relative reduction in the composite of cardiovascular death, MI, and ischemic stroke with E/S compared with placebo/simvastatin. In the IMPROVE-IT, the benefit of adding ezetimibe to statin was enhanced in patients with diabetes and in high-risk patients without diabetes.

Oyama, K, et al. studied to evaluate the relationship between baseline LDL-C above and below 70 mg/dL, and the benefit of adding ezetimibe to statin in patients post-ACS [121]. The ezetimibe/simvastatin reduced the primary endpoint by 6–8% regardless of the baseline LDL-C as compared with placebo/simvastatin, suggesting that adding ezetimibe to statin consistently reduced the risk for cardiovascular events in post-ACS patients irrespective of baseline LDL-C values.

## 11. Conclusions

Molecular biological mechanisms for the statin residual CVD risk include an increase of atherogenic lipoproteins such as small dense LDL and remnants, vascular injury and remodeling by inflammatory cytokines, and disturbed reverse cholesterol transport. PPARα agonists improve atherogenic lipoproteins, reverse the cholesterol transport system, and have vascular protective effects, such as an anti-inflammatory effect and the reduction of the oxidative state. Ezetimibe reduces intestinal cholesterol absorption and serum plant sterols, which are increased by statins and are atherogenic, possibly contributing to reduce the statin residual CVD risk. 

## Figures and Tables

**Figure 1 ijms-23-03418-f001:**
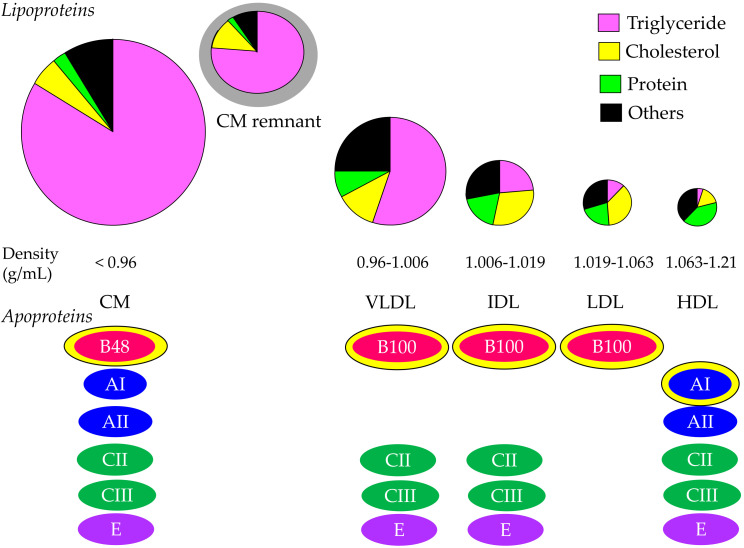
Composition and apoproteins included in each lipoprotein. Yellow–bordered apoproteins indicate that they are the main apoproteins in each lipoprotein. CM, chylomicron; HDL, high–density lipoprotein; IDL, intermediate–density lipoprotein; LDL, low–density lipoprotein; VLDL, very–low–density lipoprotein.

**Figure 2 ijms-23-03418-f002:**
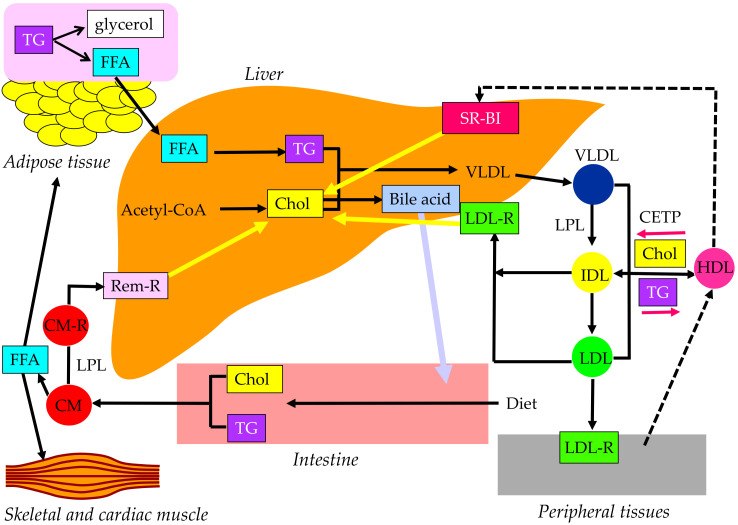
Metabolism of lipoproteins. CETP, cholesterol ester transfer protein; CM, chylomicron; CM-R, chylomicron receptor; Chol, cholesterol; FFA, free fatty acid; HDL, high-density lipoprotein; IDL, intermediate-density lipoprotein; LDL, low-density lipoprotein; LDL-R, low-density lipoprotein receptor; LPL, lipoprotein lipase; Rem-R, remnant receptor; SR-BI, scavenge receptor BI; TG, triglyceride; VLDL, very-low-density lipoprotein.

**Figure 3 ijms-23-03418-f003:**
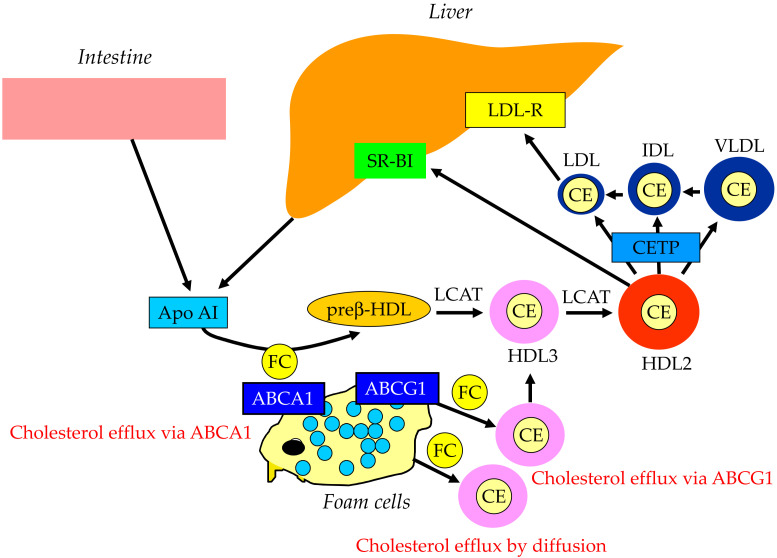
Reverse cholesterol transport. ABCA1, ATP-binding cassette transporter A1; ABCG1, ATP-binding cassette transporter G1; CE, cholesterol ester; CETP, cholesterol ester transfer protein; FC, free cholesterol; HDL, high-density lipoprotein; IDL, intermediate-density lipoprotein; LDL, low-density lipoprotein; LDL-R, low-density lipoprotein receptor; LCAT, lecithin–cholesterol acyltransferase; SR-BI, scavenge receptor BI; VLDL, very-low-density lipoprotein.

**Figure 4 ijms-23-03418-f004:**
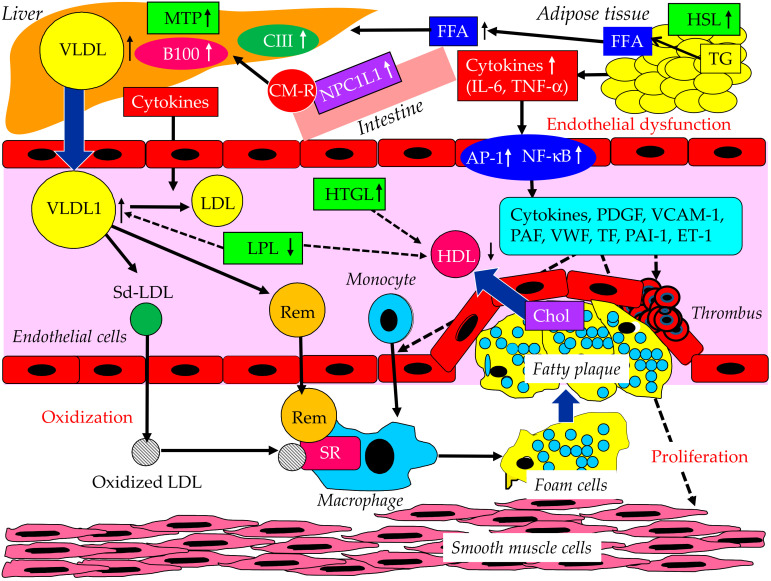
Molecular and biological mechanisms for the development of atherosclerosis by high TG and/or low HDL-C in insulin resistance. AP-1, activator protein 1; Chol, cholesterol; CM-R, chylomicron remnant; ET-1, endothelin-1; FFA, free fatty acid; HDL, high-density lipoprotein; HSL, hormone-sensitive lipase; HTGL, hepatic triglyceride lipase; LDL, low-density lipoprotein; LPL, lipoprotein lipase; MTP, microsomal triglyceride transfer protein; Sd-LDL, small dense LDL; NF-κB, nuclear factor kappa B; NPC1L1,Niemann–Pick C1-Like 1; PAF, platelet-activating factor; PAI-1, plasminogen activator inhibitor-1; PDGF, platelet-derived growth factor; SR, scavenger receptor; Rem, remnant; TF, tissue factor; VCAM-1, vascular cell adhesion molecule-1; VLDL, very-low-density lipoprotein; VWF, von Willebrand factor.

**Figure 5 ijms-23-03418-f005:**
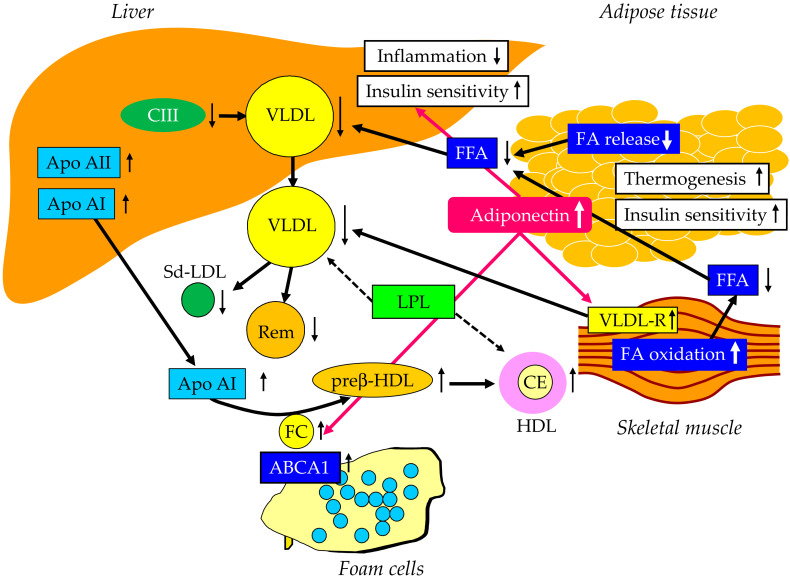
PPARα agonist-induced improvements of lipid metabolism. ABCA1, ATP-binding cassette transporter A1; CE, cholesterol ester; FC, free cholesterol; FFA, free fatty acid; HDL, high-density lipoprotein; IDL, intermediate-density lipoprotein; LPL, lipoprotein lipase; Sd-LDL, small dense LDL; Rem, remnant; VLDL, very-low-density lipoprotein; VLDL-R, very-low-density lipoprotein receptor.

**Figure 6 ijms-23-03418-f006:**
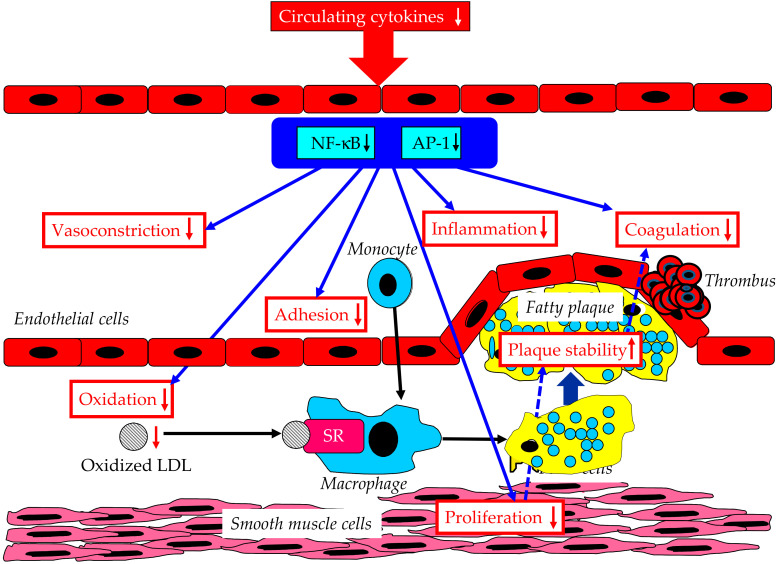
Beneficial effects of PPARα agonists on vascular integrity. AP-1, activator protein 1; NF-κB, nuclear factor kappa B; SR, scavenger receptor.

**Table 1 ijms-23-03418-t001:** Randomized, double blind, placebo-controlled trials to study the effect of statins on development of caridovascular events.

	Primary Prevention					Secondary Prevention		
Trials	WOSCOPS [2]	ASCOT-LLA [3]	AFCAPS/ TexCAPS [4]	CARDS [5]	JUPITER [6]	4S [1]	LIPID [7]	CARE [8]
Used statins and daily dose	pravastatin 40 mg	atorvastatin 10 mg	lovastatin 20–40 mg	atorvastatin 10 mg	rosuvastatin 20 mg	simvastatin 20 mg	pravastatin 40 mg	pravastatin 40 mg
Follow-up (years)	4.9	3.3	5.2	4	1.9	5.4	6.1	5
Risk reduction of cardiovascular events (statins vs. placebo, %)	31	21	37	37	44	35	29	24
Risk reduction of cardiovascular events/year (%)	6.3	6.4	7.1	9.3	23.2	6.5	4.8	4.8
Residual CVD risk (%)	69	79	63	63	56	65	71	76
Residual CVD risk/year (%)	93.7	93.6	92.9	90.8	76.8	93.5	95.2	95.2
Achieved serum lipid levels								
LDL-C (mg/dL)	142	90	115	81	55	122	113	97
HDL-C (mg/dL)	46	51	39	49	50	49	38	NA
TG (mg/dL)	142	114	143	143	99	119	126	NA
Baseline data								
BMI (kg/m^2^)	26	28.6	27.1 (men) 26.4 (women)	28.7	28.4	26	BMI > 30kg/m^2^, 18%	28
Prevalence of diabetes (%)	1	24.3	6.8	100	0	5	9	14

Gray columns were considered to be the statin residual metabolic cardiovascular risk. BMI, body mass index; HDL-C, high-density lipoprotein-cholesterol; LDL-C, low-density lipoprotein-cholesterol; NA, not available; TG, triglyceride.

**Table 2 ijms-23-03418-t002:** R+A1:D11andomized, double blind, placebo-controlled trials to study the effect of high-dose and usual-dose of statins on the development of cardiovascular events.

Trials	PROVE-IT TIMI22 [9]	IDEAL [10]	TNT [11]
Used statins and daily dose	pravastatin 40 mg vs. atorvastatin 80 mg	simvastatin 20 mg vs. atorvastatin 80 mg	atorvastatin 10 mg vs. atorvastatin 80 mg
Follow-up (years)	2	4.8	4.9
Risk reduction of cardiovascular events (high-dose vs. usual-dose, %)	16	13	22
Risk reduction of cardiovascular events/year (%)	8	2.7	4.5
Residual CVD risk (%)	84	87	78
Residual CVD risk/year (%)	92	97.3	95.5
Achieved serum lipid levels			
LDL-C	95 vs. 62	99.8 vs. 80	101 vs. 77
HDL-C	42 vs. 40	50.6 vs. 50.1	apprpximately 48 vs. 49
TG	175.8 vs. 136.2	137.2 vs. 118.5	apprpximately 140 vs. 160
Baseline data			
BMI (kg/m^2^)	29.6 vs. 29.6	27.3 vs. 27.3	28.6 vs. 28.4
Prevalence of diabetes (%)	18 vs. 18	12 vs. 12	15 vs. 15

Gray columns were considered to be the statin residual metabolic cardiovascular risk. BMI, body mass index; HDL-C, high-density lipoprotein-cholesterol; LDL-C, low-density lipoprotein-cholesterol; NA, not available; TG, triglyceride.

## Data Availability

Not applicable.

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
