# Peer review of "Molecular Biological and Clinical Understanding of the Statin Residual Cardiovascular Disease Risk and Peroxisome Proliferator-Activated Receptor Alpha Agonists and Ezetimibe for Its Treatment"

_ijms, 2022, doi:10.3390/ijms23073418_

Round 1
Reviewer 1 Report
Yanai H et al presented a comprehensive review entitled: “Molecular Biological and Clinical Understanding of the Statin Residual Cardiovascular Disease Risk and Peroxisome Proliferator-Activated Receptor Alpha Agonists for its Treatment”. This is very interesting review. In my view, several issues should be additionally emphasized and focused.
There is a lot of mechanistic, “molecular biological”, data in this nicely written and valuable comprehensive review, whereas the “clinical understanding” is much less represented. Since clinical data are obviously needed to put mechanistic data (that are sometimes indeed hypotheses) into applicable perspective I would recommend adding following short paragraphs:
- Current views on estimation/calculation of residual cardiovascular risk in patients treated with statins (LDL cholesterol vs. apoB vs non-HDL-cholesterol). How management of residual cardiovascular risk addressed in current international guidelines? The following recent study could be mentioned and commented: Johannesen CDL, Mortensen MB, Langsted A, Nordestgaard BG. Apolipoprotein B and non-HDL cholesterol better reflect residual risk than LDL cholesterol in statin-treated patients. J Am Coll Cardiol 2021;77:1439–50.
- Current views on clinical approach(es) for reducing residual cardiovascular risk: 1) intensification of statin therapy with aim to reach target values; 2) improvement of lifestyle; 3) incorporation of fibrates in particular clinical scenarios (high tryglicerides, low HDL, metabolic syndrome...)
- Brief review of data from clinical studies proving/suggesting that treatment of inflammation, coagulation and platelets could decrease residual cardiovascular risk.
- A brief review (one paragraph) of clinical studies (RCT and real-world data) exploring the role of fibrates in reduction of cardiovascular events in patients with or without metabolic syndrome. There is plethora of such studies and message from these studies would be very informative. The newly evolving evidence generally supports the value of fibrates in cardiovascular risk management and the need for well-designed studies focused on subjects with metabolic syndrome or atherogenic dyslipidemia in primary and secondary prevention i.e. reduction of residual cardiovascular risk…
Author Response
The List of Modification
Reviewer 1
- According to the suggestion “There is a lot of mechanistic, “molecular biological”, data in this nicely written and valuable comprehensive review, whereas the “clinical understanding” is much less represented. Since clinical data are obviously needed to put mechanistic data (that are sometimes indeed hypotheses) into applicable perspective I would recommend adding following short paragraphs:
Current views on estimation/calculation of residual cardiovascular risk in patients treated with statins (LDL cholesterol vs. apoB vs non-HDL-cholesterol). How management of residual cardiovascular risk addressed in current international guidelines? The following recent study could be mentioned and commented: Johannesen CDL, Mortensen MB, Langsted A, Nordestgaard BG. Apolipoprotein B and non-HDL cholesterol better reflect residual risk than LDL cholesterol in statin-treated patients. J Am Coll Cardiol 2021;77:1439–50.
Current views on clinical approach(es) for reducing residual cardiovascular risk: 1) intensification of statin therapy with aim to reach target values; 2) improvement of lifestyle; 3) incorporation of fibrates in particular clinical scenarios (high tryglicerides, low HDL, metabolic syndrome...)”
We added the following sentences by citing ref. 63 (Johannesen, C.D.L.; et al. Apolipoprotein B and Non-HDL Cholesterol Better Reflect Residual Risk Than LDL Cholesterol in Statin-Treated Patients. J. Am. Coll. Cardiol. 2021, 77, 1439-1450.)
- Current Views on Estimation of Residual Cardiovascular Risk in Patients Treated with Statins
In the guidelines on the management of CVD, LDL-C remains the primary target while apo B and non-HDL-C can be secondary targets. Atherogenic TG-rich lipoproteins such VLDL and IDL, and LDL have apo B-100. Serum level of non-HDL-C is calculated by subtracting HDL-C from total cholesterol, therefore, non-HDL also includes atherogenic TG-rich lipoproteins (VLDL, IDL and remnant) and LDL. Johannesen CDL, et al. studied to determine if elevated apo B and/or non-HDL-C are superior to elevated LDL-C in identifying statin-treated patients at residual risk of all-cause mortality and MI. In total, 13,015 statin-treated patients from the Copenhagen General Population Study were included with 8 years median follow-up. High levels of apo B and non-HDL-C were associated with increased risk of all-cause mortality and MI, whereas no such associations were found for high LDL-C, suggesting that elevated apo B and non-HDL-C, but not LDL-C, are associated with residual risk of all-cause mortality and MI in statin-treated patients [63].
- Current Views on Clinical Approaches Reducing the Statin Residual CVD Risk
To reduce the statin residual CVD risk, the following approaches should be needed. 1) intensification of statin therapy with aim to reach target values; 2) improvement of lifestyle; 3) incorporation of peroxisome proliferator-activated receptor alpha (PPARa) agonists and/or NPC1L1 inhibitor (ezetimibe) in particular clinical scenarios such as high TG, low HDL-C, obesity and metabolic syndrome.
- According to the suggestion “Brief review of data from clinical studies proving/suggesting that treatment of inflammation, coagulation and platelets could decrease residual cardiovascular risk.”
We added the following sentences by citing ref. 89 and ref. 105.
A RCT of canakinumab, a therapeutic monoclonal antibody targeting interleu-kin-1β, involving 10,061 patients with previous MI and a high-sensitivity C-reactive (hs-CRP) level of 2 mg or more per liter was performed [89]. Canakinumab did not re-duce lipid levels from baseline. At a median follow-up of 3.7 years, the 150-mg dose met the prespecified multiplicity-adjusted threshold for statistical significance for the primary end point and the secondary end point that additionally included hospitalization for unstable angina that led to urgent revascularization. Anti-inflammatory therapy targeting the interleukin-1β innate immunity pathway with canakinumab led to a significantly lower rate of recurrent cardiovascular events than placebo, independent of lipid-level lowering. This result proposes inflammatory changes are responsible for maintenance of the atherosclerotic process and may underlie vascular complications.
In the meta-analysis to analyze the risks and benefits of low-dose aspirin in pa-tients with type 2 diabetes without cardiovascular conditions, the low-dose aspirin use was associated with reduced risk for major adverse cardiovascular events (MACE) in the moderate/high-risk group [odds ratio (OR), 0.88; 95%CI, 0.80 to 0.97) [105]. Based on the within-subgroups results, the baseline cardiovascular risk does not modify the effect of aspirin therapy. This study suggests that anti-platelet activity can reduce CVD in the high risk subjects such as type 2 diabetic patients.
- According to the suggestion “A brief review (one paragraph) of clinical studies (RCT and real-world data) exploring the role of fibrates in reduction of cardiovascular events in patients with or without metabolic syndrome. There is plethora of such studies and message from these studies would be very informative. The newly evolving evidence generally supports the value of fibrates in cardiovascular risk management and the need for well-designed studies focused on subjects with metabolic syndrome or atherogenic dyslipidemia in primary and secondary prevention i.e. reduction of residual cardiovascular risk…”
We added the following paragraph by citing ref. 106-111.
- Effects of PPARα agonists on CVD
In the meta-analysis to investigate the influence of fibrates on vascular risk reduction in subjects with atherogenic dyslipidemia [106], compared to placebo, the greatest benefit with fibrate treatment was seen in high TG subjects, fibrate therapy reduced risk of vascular events [relative risk (RR), 0.75; 95%CI, 0.65 to 0.86; P < 0.001); and in subjects with both high TG and low HDL-C (RR, 0.71; 95%CI, 0.62 to 0.82; P < 0.001). Less benefit was noted in low HDL-C subjects (RR, 0.84; 95%CI, 0.77 to 0.91; P < 0.001). Among subjects with neither high TG nor low HDL-C, fibrate therapy did not reduce subsequent vascular events.
Two major trials with fenofibrate in patients with type 2 diabetes have failed to reduce MACE. The Fenofibrate Intervention and Event Lowering in Diabetes (FIELD) study randomized diabetic patients with TG 154 mg/dL, HDL-C 43 mg/dL, and LDL-C 119mg/dL to placebo and fenofibrate over a 5-year study period, and did not show a significant reduction in the primary combined endpoint for CVD between study groups [107]. However, the post hoc analysis showed that patients with elevated TG (> 200 mg/dL) or low HDL-C (< 40 mg/dL in men and < 50 mg/d in women) derived greater CV risk reduction with fenofibrate [108]. The Action to Control Cardiovascular Risk in Diabetes lipid trial (ACCORD Lipid) reported no effects on the primary composite outcome of fenofibrate added to simvastatin [109]. However, in a prespecified subgroup analysis, there was a trend towards benefit of fenofibrate in patients with TG level of ≥ 204 mg/dL or HDL-C of ≤ 34 mg/dL [109]. The post hoc analyses of the fibrate trials have shown reductions in CV events in subgroups with features of the metabolic syndrome, including overweight participants with high TG and low HDL-C [110,111].
The newly evolving evidence generally supports the value of fibrates in CV risk management. However, to confirm the usefulness of PPARα agonists for the statin residual CVD risk, the well-designed studies focused on subjects with metabolic syndrome or atherogenic dyslipidemia in primary and secondary prevention should be needed in the future.
Reviewer 2 Report
The authors discussed an important clinical issue that is encountered in the treatment of hypercholesterolemic patients with statins, i.e. the residual risk for development of cardiovascular events even after treatment with high dose statins. The authors compare several randomized, double blind, placebo-controlled trials, highlight the observed residual risks, define additional risk factors like obesity, inflammation and insulin resistance and discuss the potential of treatment with PPARα agonists. The paper is well written.
Comments
Comparing published studies introduces the problem that the studies have been performed in different countries with different ethnicities and genders, performed by different institutes involving different protocols, patients and analytical methods. The authors dealt with this correctly, presenting data from low to high and interpreting them as indications. Only figure 4 presents a problem. Values from different studies are discussed in terms of being lower and higher. Large standard deviations are observed in figure 4. Interpretation of data in terms of higher or lower is not allowed without statistical analysis.
The studies have short follow up time being 2 to 6 years. Maybe the interpretation of residual risk should be adjusted to the follow up time.
In figure 2 a large yellow arrow is shown indicating a cholesterol flux from the intestine to the liver independent of chylomicron formation. Most likely the arrow should be reversed indicating biliary cholesterol secretion.
In figure 3 the foam cells appear to release CE via SR-B1. SR-B1 is an uptake protein, not a release protein.
Tables 1 and 2: Plasma lipids have no units
In the text many times the word “after” is used where “alter” seems to be appropriate.
Interestingly, the authors focus on the alternative risk factors obesity, insulin resistance and inflammation as potential tools for treatment. They do not mention cholesterol absorption as a risk factors. Many other scientific papers specifically discuss cholesterol absorption as an important target for additional treatment. Also plant sterols are being discussed as being potentially atherogenic. Ezetimibe treatment reduces absorption of cholesterol and plant sterols. Ezetimibe treatment lowers LDL-cholesterol when added to statin treatment. An important observation is that ezetimibe treatment reduces the residual risk factor much more than may be expected from its effect on LDL-cholesterol (Giugliano, Circulation 2018; Oyama, J Am Coll Cardiol. 2021).
The authors spend many pages on the discussion of the additional risk factors. This section maybe shortened and a new section on cholesterol absorption added. This may strengthen the paper.
Author Response
The List of Modification
Reviewer 2
- According to the suggestion “Only figure 4 presents a problem. Values from different studies are discussed in terms of being lower and higher. Large standard deviations are observed in figure 4. Interpretation of data in terms of higher or lower is not allowed without statistical analysis.”
We deleted figure 4 and the description related with figure 4.
- According the suggestion “The studies have short follow up time being 2 to 6 years. Maybe the interpretation of residual risk should be adjusted to the follow up time.”
We remade Table 1 and 2 by adding the column “Residual CVD risk / year (%)”, and added the following sentences.
In these studies, the annual residual CVD risk was surprisingly high with 76.8-95.2%.
In these studies, the annual residual CVD risk was extremely high with 92-97.3%.
- According to the suggestion “In figure 2 a large yellow arrow is shown indicating a cholesterol flux from the intestine to the liver independent of chylomicron formation. Most likely the arrow should be reversed indicating biliary cholesterol secretion.”
We corrected figure 2.
- According to the suggestion “In figure 3 the foam cells appear to release CE via SR-B1. SR-B1 is an uptake protein, not a release protein.”
We corrected figure 3.
- According to the suggestion “Tables 1 and 2: Plasma lipids have no units”
We added units in Tables 1 and 2.
- According to the suggestion “In the text many times the word “after” is used where “alter” seems to be appropriate.”
We corrected.
- According to the suggestion “Interestingly, the authors focus on the alternative risk factors obesity, insulin resistance and inflammation as potential tools for treatment. They do not mention cholesterol absorption as a risk factors. Many other scientific papers specifically discuss cholesterol absorption as an important target for additional treatment. Also plant sterols are being discussed as being potentially atherogenic. Ezetimibe treatment reduces absorption of cholesterol and plant sterols. Ezetimibe treatment lowers LDL-cholesterol when added to statin treatment. An important observation is that ezetimibe treatment reduces the residual risk factor much more than may be expected from its effect on LDL-cholesterol (Giugliano, Circulation 2018; Oyama, J Am Coll Cardiol. 2021). The authors spend many pages on the discussion of the additional risk factors. This section maybe shortened and a new section on cholesterol absorption added. This may strengthen the paper.”
We added the following two more paragraphs by citing ref. 27, 28, 112-115.
We also changed the title to “Molecular Biological and Clinical Understanding of the Statin Residual Cardiovascular Disease Risk and Peroxisome Proliferator-Activated Receptor Alpha Agonists and Ezetimibe for its Treatment”.
- A Significance of Cholesterol Absorption and Plant Sterol for the Statin Residual CVD Risk
Insulin resistance enhances intestinal cholesterol absorption attributable to changes in NPC1L1 [27]. Intestinal cholesterol absorption in obese people is more than twice that in non-obese people [28]. Diabetic patients had more NPC1L1 mRNA than the control subjects (P < 0.02) [112].
Miettinen, T.A., et al. studied changes in serum cholestanol and plant sterols (indexes of cholesterol absorption) and cholesterol precursors (indexes of cholesterol synthesis) in response to cholesterol reduction by way of 1 year's treatment with atorvastatin and simvastatin treatments in patients with CHD. Plant sterol concentrations were gradually increased by atorvastatin, but decreased initially by simvastatin. Effective inhibition of cholesterol synthesis and subsequent reduction in serum cholesterol levels by statins lead to increases in serum plant sterol levels, probably as a result of reduced biliary secretion and enhanced absorption of these sterols [113]. Because serum plant sterols have been claimed to be involved in the early development of atherosclerosis, elevation of serum plant sterols can be the statin residual risk.
Miettinen, T.A., et al. studied to investigate whether baseline serum cholestanol : cholesterol ratio, which is negatively related to cholesterol synthesis, could predict reduction of coronary events in the 4S [114]. With increasing baseline quarter of cholestanol distribution the reduction in relative risk increased gradually from 0.623 (95%CI, 0.395 to 0.982) to 1.166 (0.791 to 1.72). The risk of recurrence of major coronary events increased 2.2-fold (P < 0.01) by multiple logistic regression analysis between the lowest and highest quarter of cholestanol. Measurement of serum cholestanol concentration revealed a subgroup of patients with CHD in whom coronary events were not reduced by simvastatin treatment. This study suggests that cholesterol absorption can be also the statin-residual risk.
In mildly hypercholesterolemic type 2 diabetic patients, the only metabolic parameter differentiating CHD patients from non-CHD ones was significantly higher cholesterol absorption efficiency in the coronary patients, suggesting that a significance of cholesterol absorption for atherogenesis [115].
- Effects of Ezetimibe on the Statin Residual CVD Risk
Ezetimibe, an inhibitor of intestinal cholesterol absorption, can decrease LDL-C, TG and apo B levels and increase HDL-C levels [116]. The meta-analysis showed that the combination of statins with ezetimibe as well as high-dose statin reduces plasma ox-LDL in comparison with low-to-moderate intensity statin therapy alone [117]. Ezetimibe produced significant and progressive reductions in plasma plant sterol concentrations in patients with sitosterolemia [118]. The meta-analysis showed that ezetimibe/simvastatin produced significantly greater reductions compared with simvastatin alone in LDL-C (52.5% vs 38.0%, respectively) and CRP levels (31.0% vs 14.3%, respectively) [119]. Apart from lipid-lowering, ezetimibe may exert certain off-target actions such as anti-inflammatory, anti-atherogenic and antioxidant, contributing to a further decrease of CVD risk.
Ezetimibe treatment reduces absorption of cholesterol and plant sterols. Ezetimibe treatment lowers LDL-C when added to statin treatment. Ezetimibe treatment may reduce the statin residual CVD risk. In IMPROVE-IT, 18 144 patients after ACS with LDL-C 50 to 125 mg/dL were randomized to 40 mg ezetimibe/simvastatin (E/S) or 40 mg placebo/simvastatin [120]. In patients with diabetes, E/S reduced the 7-year Kaplan-Meier primary end point event rate by 5.5%. The largest relative reductions in patients with diabetes were in MI (24%) and ischemic stroke (39%). Among patients without diabetes, those with a high risk score experienced a significant (18%) relative reduction in the composite of cardiovascular death, MI, and ischemic stroke with E/S compared with placebo/simvastatin. In IMPROVE-IT, the benefit of adding ezetimibe to statin was enhanced in patients with diabetes and in high-risk patients without diabetes.
Oyama, K, et al. studied to evaluate the relationship between baseline LDL-C above and below 70 mg/dL and the benefit of adding ezetimibe to statin in patients post-ACS [121]. The ezetimibe/simvastatin reduced primary endpoint by 6-8% regardless of baseline LDL-C as compared with placebo/simvastatin, suggesting that adding ezetimibe to statin consistently reduced the risk for cardiovascular events in post-ACS patients irrespective of baseline LDL-C values.
Round 2
Reviewer 2 Report
The authors responded well to this reviewer comments. Only, the English grammar to the added paragraphs must be checked
This manuscript is a resubmission of an earlier submission. The following is a list of the peer review reports and author responses from that submission.